# Morphological Dependence of Breast Cancer Cell Responses to Doxorubicin on Micropatterned Surfaces

**DOI:** 10.3390/polym14142761

**Published:** 2022-07-06

**Authors:** Jing Zheng, Rui Sun, Huajian Chen, Tianjiao Zeng, Toru Yoshitomi, Naoki Kawazoe, Yingnan Yang, Guoping Chen

**Affiliations:** 1Research Center for Functional Materials, National Institute for Materials Science, 1-1 Namiki, Tsukuba 305-0044, Japan; mr.oldcat@gmail.com (J.Z.); sun.rui.nims@gmail.com (R.S.); chenhuajian1118@gmail.com (H.C.); zeng.tianjiao@nims.go.jp (T.Z.); yoshitomi.toru@nims.go.jp (T.Y.); kawazoe.naoki@nims.go.jp (N.K.); 2Department of Materials Science and Engineering, Graduate School of Pure and Applied Sciences, University of Tsukuba, 1-1-1 Tennodai, Tsukuba 305-8577, Japan; 3Graduate School of Life and Environmental Science, University of Tsukuba, 1-1-1 Tennodai, Tsukuba 305-8572, Japan; yo.innan.fu@u.tsukuba.ac.jp

**Keywords:** micropatterned surface, micropattern, cell morphology, spreading area, elongation, breast cancer cells, MDR, chemoresistance, doxorubicin

## Abstract

Cell morphology has been widely investigated for its influence on the functions of normal cells. However, the influence of cell morphology on cancer cell resistance to anti-cancer drugs remains unclear. In this study, micropatterned surfaces were prepared and used to control the spreading area and elongation of human breast cancer cell line. The influences of cell adhesion area and elongation on resistance to doxorubicin were investigated. The percentage of apoptotic breast cancer cells decreased with cell spreading area, while did not change with cell elongation. Large breast cancer cells had higher resistance to doxorubicin, better assembled actin filaments, higher DNA synthesis activity and higher expression of P-glycoprotein than small breast cancer cells. The results suggested that the morphology of breast cancer cells could affect their resistance to doxorubicin. The influence was correlated with cytoskeletal organization, DNA synthesis activity and P-glycoprotein expression.

## 1. Introduction

Chemoresistance is one of the major obstacles of cancer therapy [1,2]. Chemoresistance is strongly related with the tumor microenvironment that is composed of specific extracellular matrices, signal molecules and stroma cells [3]. Tumor microenvironment stress can assist the development of chemoresistance through influences on cancer cell survival and therapeutic response to anti-cancer drugs [4,5,6,7].

In addition to the cellular microenvironment, cell morphology has been reported to affect the functions of normal cells [8,9,10,11,12]. Cell morphology affects cell adhesion, proliferation, differentiation, cellular uptake and gene transfection [13,14,15,16,17,18,19]. Cell apoptosis can be also affected by cell morphology [20,21]. Human capillary endothelial cells with a small round morphology have a high tendency of apoptosis. Big cell size and large spreading area increase cell viability [20]. Size confinement of mesenchymal stem cells induces an apoptotic tendency, while big cell size is beneficial for osteogenesis [21].

Cell morphology affects not only the functions of normal cells, but also cancer cells. Osteosarcoma MG-63 cells change their stiffness when cell size varies [22]. Cell spreading area and cell shape of human breast carcinoma cells and lung carcinoma cells affect cellular uptake of exogenous polymer micelles [23]. Although some evidence has disclosed the influence of cancer cell morphology on cell mechanical property and uptake of exogenous agents, it is unclear how the morphology of cancer cells affects chemoresistance.

Therefore, in this study, spreading area (size) and elongation (aspect ratio) of MDA-MB-231-Luc cells (human breast cancer cell line) were controlled and their influences on residence to doxorubicin were investigated. In our previous studies, micropatterned surfaces have been prepared by using photo-reactive poly vinyl alcohol (PVA) [17,24]. The PVA-micropatterned surfaces are stable during cell culture and can be used to precisely control cell morphology by designing the micropattern structures. In this study, the micropatterning method was used to prepare micropatterned surfaces with round circle micropatterns of a diameter of 20, 30, 40 and 60 μm and elliptic micropatterns of an aspect ratio of 2:1, 4:1 and 8:1. Breast cancer cells were cultured on the micropatterned surfaces and their responses to doxorubicin were elucidated.

## 2. Materials and Methods

### 2.1. Preparation of Micropatterned Surfaces

Micropatterned surfaces with different sizes of circles (diameter of 20, 30, 40 and 60 μm) and different aspect ratios of ellipses (aspect ratio of 2:1, 4:1 and 8:1) were fabricated by photolithography on polystyrene plates using photo-reactive PVA. The photo-reactive azidophenyl-derivatized PVA was synthesized by grafting azidophenyl groups into the side chains of PVA (Wako Pure Chemical Industries, Osaka, Japan), as previously reported [8]. Polystyrene plates with a size of 2.5 × 2.5 cm^2^ were cut from Falcon^TM^ tissue culture flasks (Corning Life Sciences, Durham, NC, USA). Aqueous solution of the photo-reactive PVA (0.3 mg/mL, 0.2 mL) was coated on the central area (1.5 × 1.5 cm^2^) of polystyrene plates. After air-drying in the dark, the photo-reactive PVA-coated polystyrene plates were covered by a photomask with the designed micropatterns and UV-irradiated in a Funa-UV-linker (Funakoshi Co., Ltd., Tokyo, Japan) at a total energy of 0.25 J/cm^2^. After UV irradiation, the micropatterned polystyrene plates were washed with Milli-Q water in an ultrasonic cleaner.

### 2.2. Characterization of Micropatterned Surfaces

Phase-contrast images of the photomasks and the micropatterned surfaces were captured with an Olympus BX50 microscope (Olympus, Tokyo, Japan). The surface topography of the micropatterned surface was characterized by an MFP-3D-BIO atomic force microscope (AFM, Asylum Research Corporation, Santa Barbara, CA, USA). The AFM characterization was conducted in Milli-Q water using a contact mode with a cantilever purchased from Bruker, Camarillo, CA, USA (DNP-10, spring constant: 0.06 N/m; oscillation frequency: 12–24 kHz).

### 2.3. Coating of Fibronectin and Immunological Staining of Coated Fibronectin

Before being used for cell culture, the micropatterned surfaces were coated with fibronectin to enhance cell adhesion. The micropatterned polystyrene plates were first sterilized by soaking in a 70% ethanol aqueous solution for 30 min, followed by rinsing with Milli-Q water 3 times. A total of 0.2 mL of 20 μg/mL fibronectin (Sigma-Aldrich, St. Louis, MO, USA) in a NaHCO_3_ (pH = 8.4) aqueous solution was spread on the micropatterned surfaces. After being incubated at 37 °C for 1 h, the micropatterned plates were washed with the NaHCO_3_ aqueous solutions 3 times, and then further washed another 3 times with sterile Milli-Q water.

To characterize the fibronectin-coated micropatterned surfaces, immunological staining was performed using a mouse anti-fibronectin primary antibody (1:200 in 1% bovine serum albumin (BSA) in PBS, Santa Cruz Biotechnology, Dallas, TX, USA) and an Alexa Fluor-488 goat anti-mouse IgG (1: 1000 in PBS, Invitrogen, Waltham, MA, USA) as the secondary antibody. The stained micropatterned surfaces were observed under a fluorescence microscope.

### 2.4. Cell Culture

MDA-MB-231-Luc cells, human breast cancer cell line, were obtained from the Japanese Collection of Research Bioresources Cell Bank (Osaka, Japan). MDA-MB-231-Luc cells were cultured in the growth medium of an L-15 medium (L5520, Sigma-Aldrich, St. Louis, MO, USA) supplemented with 10% fetal bovine serum (FBS, Gibco, Grand Island, NY), 100 U-100 µg/mL penicillin-streptomycin (Sigma-Aldrich, St. Louis, MO, USA) and 4 mM L-glutamine (Sigma-Aldrich, St. Louis, MO, USA). The subcultured cells at an 80% confluence were harvested by trypsinization and re-suspended in the growth medium for cell seeding. The fibronectin-coated micropatterned plates were placed in 6 well culture plates. A total of 3 mL of growth medium was added to each well. To avoid cell leakage during cell seeding, a glass ring (a diameter of 1.5 cm) was placed on each micropatterned plate. The cell density was adjusted to 2.5 × 10^4^ cells/mL and 200 μL aliquot of the cell suspension was pipetted into each glass ring. The seeded cells were then cultured in a 5% CO_2_ incubator at 37 ℃ for cell attachment. Six h later, the glass rings were removed, and the culture medium was refreshed to remove the unattached cells. The cell seeding procedure was conducted once again to allow more MDA-MB-231-Luc cells to attach on the micropatterns. After culture for 6 h of the second cell seeding, the glass rings were removed, and the culture medium was refreshed again. After the cells were cultured on the micropatterned surfaces for 24 h after the second cell seeding, the micropatterned cells were used for the following experiments. Cell morphology was observed with an Olympus BX50 microscope.

### 2.5. TUNEL Assay

TUNEL (terminal deoxynucleotidyl transferase-mediated deoxyuridine triphosphate (dUTP) nick end labeling) assay was performed to determine the apoptosis of MDA-MB-231-Luc cells. After culture for 24 h after cell seeding, the culture medium was changed to a doxorubicin-containing medium. The doxorubicin-containing medium was prepared by adding doxorubicin hydrochloride (Sigma-Aldrich, St. Louis, MO, USA) into the growth medium. The doxorubicin hydrochloride was first dissolved in sterilized Milli-Q water at a concentration of 5 mg/mL. After filter-sterilization, the concentrated doxorubicin solution was added to the growth medium until the final concentration reached 1 μg/mL. After 3 days culture in the doxorubicin-containing medium, the cells were washed with PBS 3 times and then fixed with PBS containing 4% paraformaldehyde (FUJIFILM Wako Pure Chemical, Osaka, Japan) for 30 min. The TUNEL assay was performed using an in situ apoptosis detection kit (Takara, Shiga, Japan). Briefly, the fixed cells were treated by the permeabilization buffer for 5 min on ice, followed by incubation with terminal deoxynucleotidyl transferase enzymes and fluorescein-dUTP at 37 ℃ for 1.5 h to label the DNA fragments from apoptotic cells. Then, the reaction was terminated by washing with PBS. Finally, the nuclei were stained with Hoechst 33,258 (dilute in PBS at 1:1000, FUJIFILM Wako Pure Chemical, Osaka, Japan) in the dark for 10 min. The samples were then observed under a fluorescence microscope with a DP-70 CCD camera. The number of apoptotic cells was counted from the stained samples. The percentage of apoptotic cells was calculated by dividing the number of apoptotic cells with the total cell number of each micropatterned plate. The cells cultured on the micropatterned surfaces in the growth medium without doxorubicin for 3 days were also stained as a control. Three independent samples (more than 200 cells) were used for the analysis to calculate means and standard deviations.

### 2.6. Staining of Actin Filaments

After the cells were cultured for 24 h after cell seeding, the cells were fixed with PBS containing 4% paraformaldehyde for 10 min and washed with PBS 3 times. Then, the cells were permeabilized by treatment with 1% Triton^TM^ X-100 (Sigma-Aldrich, St. Louis, MO, USA), followed by blocking with 2% bovine serum albumin (BSA, FUJIFILM Wako Pure Chemical, Osaka, Japan) in PBS. After washing with PBS 3 times, the cells were stained with Alexa Fluor-488 phalloidin (dilute in PBS at 1:40, Invitrogen, Waltham, MA, USA) for 20 min at room temperature (avoid light). The stained cells were washed with PBS and stained with Hoechst 33,258 (dilute in PBS at 1:1000, FUJIFILM Wako Pure Chemical, Osaka, Japan) for 10 min to stain cell nuclei. Fluorescence images of the stained cells were obtained using a fluorescence microscope with a DP-70 CCD camera.

### 2.7. Analysis of DNA Synthesis Activity by BrdU Staining

To investigate the DNA synthesis activity of the micropatterned cells, 5-bromo-2′-deoxyuridine (BrdU, Abcam, Cambridge, UK) was used. The BrdU was first dissolved in sterilized MilliQ water at a concentration of 10 mM, followed by filter sterilization. The BrdU labeling medium containing 10 μM BrdU was prepared by diluting the 10 mM BrdU concentrated solution with the growth medium. After the cells were cultured on the micropatterned surfaces for 24 h, the culture medium was changed to the BrdU labeling medium and the cells were further cultured in the BrdU labeling medium for 24 h. Then, the cells were fixed with 70% ethanol for 30 min and treated with 2 M HCl for 30 min. After washing with PBS 3 times, the cells were permeabilized with 1% Triton TM X-100 for 10 min and washed with PBS again. Two percent BSA in PBST (PBS+ 0.1% Tween 20) was then used for blocking for 30 min. After washing with PBS 3 times, the cells were incubated with the monoclonal mouse anti-BrdU primary antibody (1: 200 in 2% BSA in PBST, Abcam, Cambridge, UK) for 1.5 h, followed by incubation with the Alexa Fluor-488 donkey anti-mouse IgG antibody (1:1000 in 2% BSA in PBST, Thermo Fisher, Waltham, MA, USA) in the dark for 1 h. The stained samples were then washed with PBS 3 times and the nuclei were stained with Hoechst 33,258 in the dark for 10 min. Fluorescence images were obtained by a fluorescence microscope. The number of BrdU positively stained cells were counted and the percentage of BrdU positively stained cells to the total cell number was calculated. Three independent samples (more than 200 cells) were used for the analysis.

### 2.8. Immunofluorescent Staining of P Glycoprotein

After the cells were cultured for 24 h after cell seeding, the cells were fixed with PBS containing 4% paraformaldehyde for 10 min and washed with PBS 3 times. Two percent BSA in PBST was then used for blocking for 30 min. After washing 3 times with PBS, the cells were incubated with the Anti-P Glycoprotein antibody (1:100 dilution in 2% BSA in PBST, Abcam, Cambridge, UK) for 1 h at room temperature, followed with incubation with the Alexa Fluor-488 goat anti-rabbit IgG antibody (1:500 in 2% BSA in PBST, Thermo Fisher, Waltham, MA, USA) at room temperature for 1 h in the dark. After 3 times PBS washing, the nuclei were stained with Hoechst 33,258 in the dark for 10 min. The fluorescence images of the stained cells were obtained by a fluorescence microscope.

### 2.9. Statistical Analysis

A one-way analysis of variance (ANOVA) with Tukey’s post hoc test for multiple comparisons was performed for statistical analysis. The quantitative data were presented as the means ± standard deviations (SDs). The results were considered significantly different when the *p* value was less than 0.05.

## 3. Results and Discussions

### 3.1. Preparation and Characterization of Micropatterned Surfaces

The micropatterned surfaces were prepared by a photolithographic method. At first, photo-reactive PVA was synthesized by introducing photo-reactive azidophenyl groups in PVA. Then, the photo-reactive PVA was micropatterned on polystyrene plates. The photomasks used for the micropatterned surfaces are shown in Figure 1a. The photomasks had round and elliptic micropatterns of different sizes and aspect ratios. The round micropatterns (aspect ratio = 1:1) had diameters of 20, 30, 40 and 60 μm. The aspect ratios of elliptic micropatterns changed from 2:1 to 4:1 and 8:1. Each row of the four micropatterns had the same area but different aspect ratios, while each column of the four micropatterns had the same aspect ratio but different areas. The micropatterned surfaces are shown in Figure 1b. The micropatterned surfaces had the same round and elliptic micropatterns as those of the photomasks.

AFM 3D images showed the pit structures of the micropatterns (Figure 2). The concave areas were the polystyrene surface, while the convex areas were the grafted PVA on the polystyrene surface. The PVA molecules were covalently grafted on the polystyrene surface. Therefore, the micropatterns should be stable during cell culture. The results indicated that the designed micropatterns were formed on the micropatterned surfaces.

### 3.2. Coating of Fibronectin and Cell Culture on Micropatterned Surfaces

The micropatterned surfaces were coated with fibronectin. Immunological staining of coated fibronectin demonstrated that fibronectin was adsorbed following the structure of the micropatterns (Figure 3a). Therefore, the stained fibronectin also reflected the structures of the micropatterns. Fibronectin was coated on the micropatterned surfaces to promote cell adhesion because fibronectin is one of the predominant cell adhesion proteins that are generally used for favorable cell adhesion [25]. Fibronectin could be adsorbed on the polystyrene areas, while not on the PVA-grafted areas. It has been reported that proteins cannot adsorb on PVA [26].

The fibronectin-coated micropatterns were used for the culture of human breast cancer cells (MDA-MB-231-Luc cells). The cells adhered on the micropatterned surfaces following the micropatterns (Figure 3b). The cells could adhere on the fibronectin-coated micropatterns, but not on the PVA areas. It has been reported that PVA without protein modification does not support cell adhesion [26]. The cancer cells adhered on the micropatterned surfaces showed the same morphology as that of the micropatterns. Therefore, the cancer cells on the same row had the same size (spreading area) but different aspect ratios (elongation). The 20, 30, 40 and 60 μm round micropatterns had the calculated areas of 314, 707, 1256 and 2826 μm^2^, respectively. The cancer cells on the same column had the same elongation but different spreading areas.

### 3.3. Doxorubicin-Induced Apoptosis of Breast Cancer Cells on Micropatterned Surfaces

After the MDA-MB-231-Luc cells were cultured on the micropatterned surfaces for 24 h, doxorubicin was added in the cell culture medium and the cells were cultured for another 3 days in the doxorubicin-containing medium. Then, the cells were evaluated by TUNEL assay to examine the apoptotic cells (Figure 4a). The green fluorescence indicated the apoptotic cells. Cell nuclei were stained blue by Hoechst 33,258. Some apoptotic cells were detected on all the micropatterns.

The number of apoptotic cells was counted and divided with the total cell number on each micropattern to calculate the percentage of apoptotic cells (Figure 4b). When the breast cancer cells had the same spreading area, but different elongations, the percentages of apoptotic cells were not significantly different. The elongation of cells with the same spreading area did not significantly affect the percentage of apoptotic cells. When cell elongation was the same, the percentage of apoptotic cells significantly decreased with the increase of spreading area. The small cancer cells had the highest percentage of apoptotic cells, while the large cancer cells had the lowest percentage of apoptotic cells. When the cells were cultured in the growth medium without doxorubicin, there were almost no apoptotic cells. The results indicated that small breast cancer cells were more sensitive to the treatment of doxorubicin than large breast cancer cells. Cell elongation did not affect the sensitivity of breast cancer cells to the treatment of doxorubicin. The results suggested that cell morphology could affect breast cancer cell resistance to doxorubicin. Large breast cancer cells had higher potential of resistance to doxorubicin than small breast cancer cells.

### 3.4. Cytoskeletal Organization of Breast Cancer Cells on Micropatterned Surfaces

To investigate the reasons why cell morphology affected the sensitivity to doxorubicin treatment, actin filaments of the breast cancer cells were stained after culture on the micropatterned surfaces for 24 h (Figure 5). Actin filament is one of the main components of cytoskeleton that localizes beneath cellular membrane. Actin filament staining showed actin filaments were more strongly stained at the peripheral regions than the central regions of the cells. Actin filaments were predominantly assembled at the cell periphery. In the body regions of the cells, actin filaments were relatively weakly stained and randomly assembled.

The breast cancer cells with the same spreading area, but different elongations had almost the same level of actin filament staining. When the spreading area increased, actin filaments were more strongly stained. The results indicated that cell spreading area could affect actin filaments assembly of breast cancer cells, while cell elongation did not. The results are consistent with previous studies [22,27]. Cancer cells have less organized actin filaments than their normal counterparts [22].

### 3.5. DNA Synthesis Activity of Breast Cancer Cells on Micropatterned Surfaces

After the cells were cultured on the micropatterned surfaces in the growth medium for 24 h, most of the cells showed single cell occupation on the micropatterns. DNA synthesis activity of single cells on the micropatterns was evaluated by BrdU staining. The cells with DNA synthesis activity were stained green (Figure 6a). Some of the micropatterned cells were positively stained. To count the total cell number, cell nuclei were stained with Hoechst 33,258.

The percentage of BrdU-positive cells was calculated by dividing the number of BrdU-positive cells with the total number of breast cancer cells on each micropattern (Figure 6b). The percentage of BrdU-positive cells was almost the same for the breast cancer cells cultured on the micropatterns having the same area. However, when cell spreading area increased, the percentage of BrdU-positive cells significantly increased. The results indicated that DNA synthesis activity increased with cell spreading area, while cell elongation did not affect DNA synthesis activity.

### 3.6. Staining of P-Glycoprotein of Breast Cancer Cells on Micropatterned Surfaces

P-glycoprotein is one of the main mediators of chemoresistance [28]. P-glycoprotein was stained after the breast cancer cells were cultured on the micropatterned surfaces for 24 h. P-glycoprotein was stained green (Figure 7). Cell nuclei were co-stained (blue). The results showed that all the cells on all the micropatterns were positively stained by the anti-P-glycoprotein antibody. The results indicated that the cells on all the micropatterns expressed P-glycoprotein to efflux the doxorubicin.

The breast cancer cells with the same spreading area, but different elongations showed almost the same level of P-glycoprotein staining. However, the staining level of P-glycoprotein increased with the spreading area when cell elongation was the same. The large cells showed the strongest staining of P-glycoprotein, and the small cells showed the weakest staining of P-glycoprotein. The results suggested that the cell spreading area could promote the expression of P-glycoprotein, but cell elongation did not affect its expression.

Cells are surrounded with their specific microenvironments in vivo. The cellular microenvironment plays an important role on cell functions [29,30,31,32,33,34]. Not only are normal somatic cells and stem cells affected by their microenvironments, but cancer cells also communicate and exchange various signals with their tumor microenvironments [4,5,6,7]. Tumor microenvironment stress can affect chemoresistance [5,6,7]. Abnormal hydrostatic pressure has been reported to enhance human breast cancer cell resistance to doxorubicin [35]. Besides the influences of cellular microenvironments, cell morphology plays a pivotal role in controlling cell functions [20,21]. In this study, cell morphology influence on chemoresistance of cancer cells was disclosed.

Many morphological features such as cell spreading, elongation, geometry, protrusion and chirality have been reported to affect cell functions [14,17,36]. The two dominant factors of cell spreading and elongation were chosen and investigated in this study. The spreading area and elongation of breast cancer cells were controlled by the micropatterned surfaces with round and elliptic micropatterns of different sizes and aspect ratios. Their influences on the resistance to doxorubicin were investigated by culturing the micropatterned breast cancer cells in doxorubicin-containing medium. The cell spreading area could affect the resistance to doxorubicin, while cell elongation did not affect the resistance to doxorubicin. Large cells facilitated doxorubicin resistance, while small cells suppressed doxorubicin resistance.

The different influences of cell spreading area and elongation on doxorubicin resistance should be correlated with their influences on cytoskeletal organization, DNA synthesis activity and P-glycoprotein expression. One of the main factors affecting drug resistance of cancer cells is the efflux transports which assist drug efflux. P-glycoprotein, multidrug resistance proteins and breast cancer resistance proteins have been reported as efflux transports which may promote the resistance of breast cancer cells to doxorubicin [37,38]. Among them, P-glycoprotein has been reported to generate the strongest drug resistance [39]. Expression of P-glycoprotein might be affected by the organization of actin filaments because highly organized actin filaments can facilitate DNA synthesis and protein synthesis [40,41]. Furthermore, the traffic of P-glycoprotein has been reported to be related to the organization of actin filaments [42]. Therefore, the stronger drug resistance found in the larger breast cancer cells might be attributed to the higher expression of P-glycoprotein and higher DNA synthesis activity as a result of better organized actin filaments. The results should provide useful information for chemoresistance and chemotherapy of breast cancer.

## 4. Conclusions

PVA-micropatterned surfaces were prepared and used to control the spreading area and elongation of the cells of breast cancer cell line. The influences of the cell spreading area and elongation on the resistance to doxorubicin were investigated. The cell spreading area could affect the resistance to doxorubicin, while cell elongation had no influence on the resistance to doxorubicin. Large breast cancer cells had higher resistance to doxorubicin than small cells. The influences were correlated with the influences of the cell spreading area and elongation on cytoskeleton structure, DNA synthesis activity and P-glycoprotein expression. The results suggested that cell morphology could affect the resistance of breast cancer cells to doxorubicin.

## Figures and Tables

**Figure 1 polymers-14-02761-f001:**
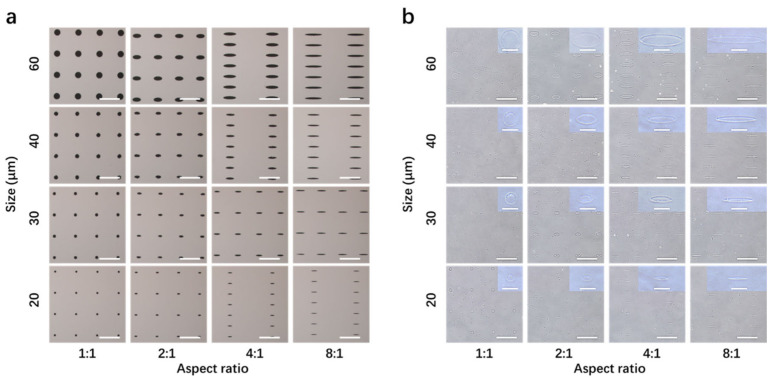
(**a**) Phase-contrast micrographs ofthe photomasks. Scale bar = 200 μm. (**b**) Phase-contrast micrographs of the micropatterned surfaces. The inserts are high-magnification photomicrographs of the micropatterns. Scale bars of the low- and high-magnification photomicrographs are 200 and 50 μm, respectively.

**Figure 2 polymers-14-02761-f002:**
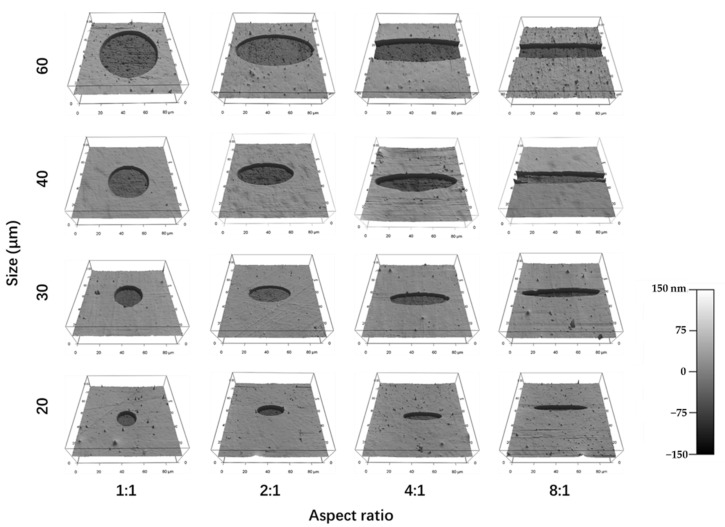
AFM 3D images of the micropatterned surfaces. The round micropatterns had a diameter of 20, 30, 40 and 60 μm. The elliptic micropatterns had an aspect ratio of 2:1, 4:1 and 8:1. The micropatterns in the same row had the same area, while the micropatterns in the same column had the same aspect ratio.

**Figure 3 polymers-14-02761-f003:**
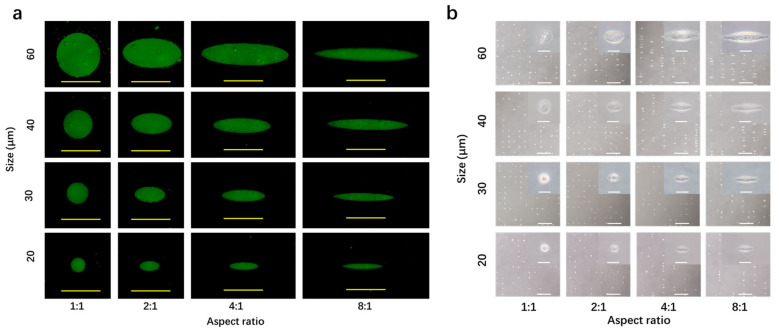
(**a**) Fluorescence photomicrographs of fibronectin-coated micropatterns. Scale bar = 50 μm. (**b**) Phase-contrast photomicrographs of breast cancer cells after culture on the micropatterned surfaces for 24 h. The inserts are high-magnification photomicrographs of the micropatterned cells. Scale bars of the low- and high-magnification photomicrographs are 500 and 50 μm, respectively.

**Figure 4 polymers-14-02761-f004:**
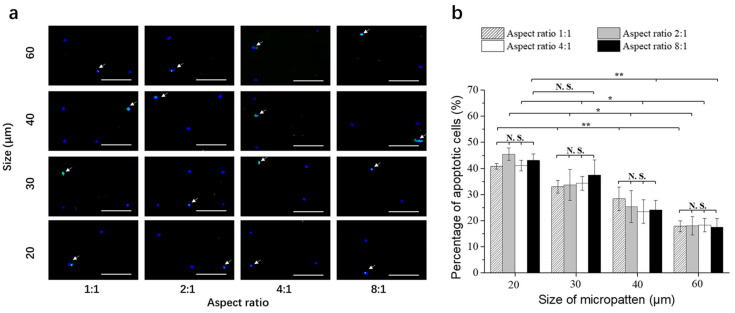
(**a**) Representative fluorescence images of the stained breast cancer cells on the micropatterned surfaces. Apoptotic cells were stained green, and nuclei were stained blue. Scale bar = 200 μm. (**b**) Percentage of apoptotic cells after culture on the micropatterned surfaces in doxorubicin-containing medium for 3 days. Data are represented as the means ± SDs (*n* = 3), N.S. means no significant difference, * *p* < 0.05, ** *p* < 0.01.

**Figure 5 polymers-14-02761-f005:**
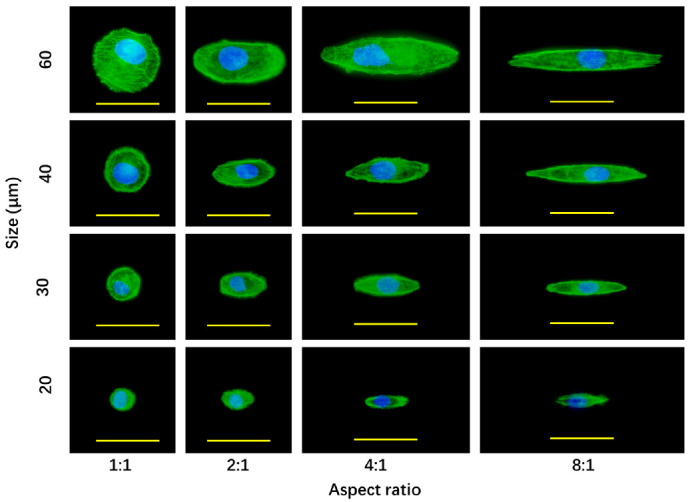
Fluorescence photomicrographs of actin filament staining of breast cancer cells after culture on the micropatterned surfaces for 24 h. Actin filaments and nuclei were stained green and blue, respectively. Scale bar = 50 μm.

**Figure 6 polymers-14-02761-f006:**
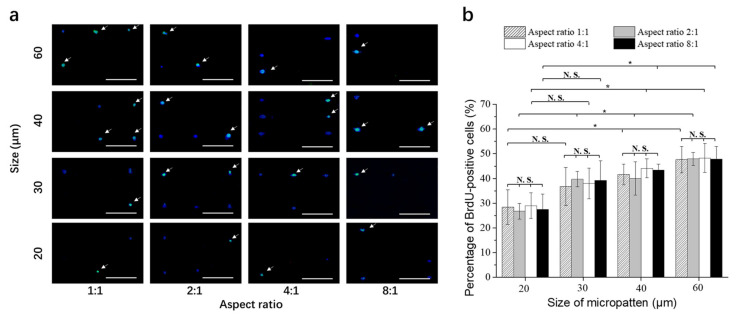
(**a**) Representative fluorescence images of BrdU staining (green) of breast cancer cells cultured on the micropatterned surfaces for 24 h. Cell nuclei were stained blue. Scale bar = 200 μm. (**b**) Percentage of BrdU positively stained breast cancer cells cultured on the micropatterned surfaces for 24 h. Data are represented as the means ± SDs (*n* = 3), N.S. means no significant difference, * *p* < 0.05.

**Figure 7 polymers-14-02761-f007:**
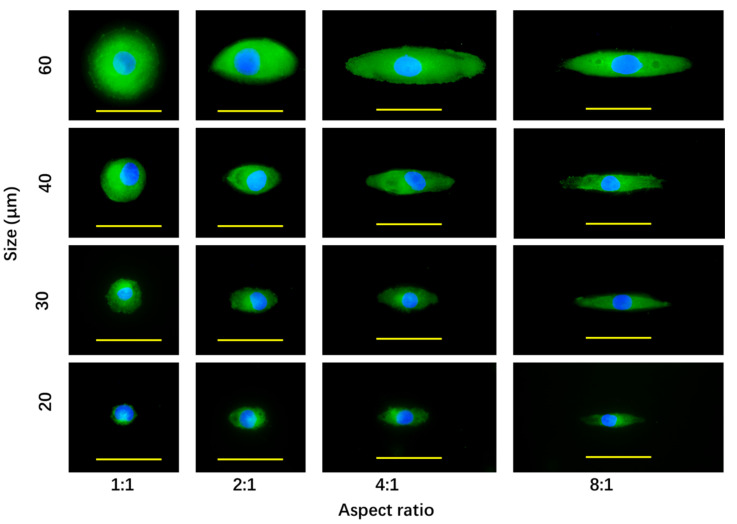
Fluorescence photomicrographs of P-glycoprotein staining of breast cancer cells after culture on the micropatterned surfaces for 24 h. P-glycoprotein and nuclei were stained green and blue, respectively. Scale bar = 50 μm.

## Data Availability

Not applicable.

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
