# Peer review of "Morphological Dependence of Breast Cancer Cell Responses to Doxorubicin on Micropatterned Surfaces"

_polymers, 2022, doi:10.3390/polym14142761_

Round 1

Reviewer 1 Report

This manuscript reports that by studying the effect of cell morphology on the resistance of cancer cells to anticancer drugs, it is finally concluded that cell morphology affects the resistance of breast cancer cells to doxorubicin. The work is interesting and would be published after minor revisions:

1. The morphology of breast cancer cells may affect their resistance to doxorubicin. It affects the expression of p-glycoprotein, DNA synthesis activity. Is there any other factor affect the resistance of breast cancer cells to doxorubicin?

2. The authors should further discuss morphological dependence of breast cancer cell responses to doxorubicin on micropatterned surfaces.

3. Are there other influencing factors besides cell spreading area and elongation?

Author Response

Thank you very much for your valuable comments and suggestions. The manuscript has been revised according to your suggestions.

Point 1: The morphology of breast cancer cells may affect their resistance to doxorubicin. It affects the expression of p-glycoprotein, DNA synthesis activity. Is there any other factor affect the resistance of breast cancer cells to doxorubicin?

Response 1: P-glycoprotein, multidrug resistance proteins and breast cancer resistance protein have been reported as efflux transports which can affect the resistance of breast cancer cells to doxorubicin. Among them, P-glycoprotein has been reported to generate the strongest drug resistance. So, in this study, we investigated the influence of cell morphology on the expression of P-glycoprotein instead of the other two. DNA synthesis activity is related to cell viability. Therefore, DNA synthesis activity was investigated in this study. This has been discussed in the revised manuscript.

Point 2: The authors should further discuss morphological dependence of breast cancer cell responses to doxorubicin on micropatterned surfaces.

Response 2: One of the main factors affecting drug resistance of cancer cells is the efflux transports which assist drug efflux. P-glycoprotein, multidrug resistance proteins and breast cancer resistance protein have been reported as efflux transports which may promote the resistance of breast cancer cells to doxorubicin. Among them, P-glycoprotein has been reported to generate the strongest drug resistance. Expression of P-glycoprotein might be affected by the organization of actin filaments because highly organized actin filaments can facilitate DNA synthesis and protein synthesis. Furthermore, the traffic of P-glycoprotein has been reported to be related to the organization of actin filaments. Therefore, the stronger drug resistance found in the larger breast cancer cells might be attributed to the higher expression of P-glycoprotein and higher DNA synthesis activity as a result of better organized actin filaments. This has been discussed in the revised manuscript and some new references have been cited to support the discussion.

Point 3: Are there other influencing factors besides cell spreading area and elongation?

Response 3: Cell spreading, elongation (aspect ratio), shape, protrusion and chirality can affect cell functions. Among these factors, cell sprading area and elongation are the dorminant ones. Therefore, cell spreading and elongation were chosen and investigated in this study. This has been discussed in the revised manuscript.

Reviewer 2 Report

Comments on Manuscript “Title: Morphological Dependence of Breast Cancer Cell Responses to Doxorubicin on Micropatterned Surfaces” by Zheng et al.

The authors studied the surface area and shape effect on MDA-MB-231 cell line w.r.t. doxorubicin resistance.

I have following recommendations before the publication of this MS, which need to be addressed in MS:

1. Figure 4-Number of apopototic cells in 4b are not significant as is evident by the cell images in 4a. So, not clear if the area really contributes to the cell kill. Important control is missing here-which is 231 cell line in growth media with and w/o doxorubicin-that is why the title is not supported by the data-should be changed.

2. Figure 6-same problem as fig. 4. Not significant cells in 6A. Author wrote in line 280-cell growth not observed-that would have been ideal experiment to check if cells proliferate with time w and w/o dox.

The paper is technically sound and well written. comments above- will make the MS better.

Author Response

Thank you very much for your valuable comments and suggestions. The manuscript has been revised according to your suggestions.

Point 1: Figure 4-Number of apopototic cells in 4b are not significant as is evident by the cell images in 4a. So, not clear if the area really contributes to the cell kill. Important control is missing here-which is 231 cell line in growth media with and w/o doxorubicin-that is why the title is not supported by the data-should be changed.

Response 1: The fluorescence images in Figure 4a are photomicrographs at a high magnification to show the fluorescene of the stained cells. The apoptotic cells were stained green and cell nuclei were stained blue. The images do not reflect the average trend of the percentage of apoptotic cells. Low magnification images can reflect the average trend of apoptotic cell percentage. However, it is difficult to see the cells if low magnication images were used. Therefore, only very limited number of cells at a high magnification are shown in Figure 4a. However, the statistic results shown in Figure 4b were obtained from 3 independent samples with more than 200 cells being checked.

The cells cultured on the micropatterned surfaces in the growth medium without doxorubicin for 3 days were also stained as a control. When the cells were cultured in the growth medium without doxorubicin, almost no apoptotic cells were detected. The information has been added in the revised manuscript.

Point 2: Figure 6-same problem as fig. 4. Not significant cells in 6A. Author wrote in line 280-cell growth not observed-that would have been ideal experiment to check if cells proliferate with time w and w/o dox.

Response 2: The images in Figure 6a were taken at a high magnification to show the fluorescene of BrdU staining. The BrdU positively stained cells are shown in green fluorescence, while cell nuclei are shown in blue. The images do not reflect the average trend of the percentage of BrdU positively stained cells. Low magnification images can reflect the average percentage of BrdU positively stained cells. However, it is difficult to see the cells if low magnication images were used. Therefore, only very limited number of cells at a high magnification are shown in Figure 6a. However, the statistic results in Figure 6b were obtained from 3 independent samples with more than 200 cells being checked.

After the cells were cultured on the micropatterned surfaces in the growth medium for 24 hours, most of the cells showed single cell occupation on the micropatterns. DNA synthesis activity of single cells on the micropatterns was evaluated by BrdU staining. We only counted single cells, not mutiple cells. For single cells, it is difficulte to evaluate cell proliferation by counting cell number. Therefore, we used BrdU staining to evaluate the DNA synthesis activity. The information has been added and some sentences have been revised in the revised manuscript.

Round 2

Reviewer 2 Report

Author revised the MS nicely as per reviewer's suggestions, including mine.